# TRANSFERABLE DEEP CLUSTERING MODEL

## ABSTRACT

Deep learning has shown remarkable success in the field of clustering recently. However, how to transfer a trained clustering model on a source domain to a target domain by leveraging the acquired knowledge to guide the clustering process remains challenging. Existing deep clustering methods often lack generalizability to new domains because they typically learn a group of fixed cluster centroids, which may not be optimal for the new domain distributions. In this paper, we propose a novel transferable deep clustering model that can automatically adapt the cluster centroids according to the distribution of data samples. Rather than learning a fixed set of centroids, our approach introduces a novel attention-based module that can adapt the centroids by measuring their relationship with samples. In addition, we theoretically show that our model is strictly more powerful than some classical clustering algorithms such as k-means or Gaussian Mixture Model (GMM). Experimental results on both synthetic and real-world datasets demonstrate the effectiveness and efficiency of our proposed transfer learning framework, which significantly improves the performance on target domain and reduces the computational cost.

## 1 INTRODUCTION

Clustering is one of the most fundamental tasks in the field of data mining and machine learning that aims at uncovering the inherent patterns and structures in data, providing valuable insights in diverse applications. In recent years, deep clustering models (Min et al., 2018; Zhou et al., 2022; Ren et al., 2022) have emerged as a major trend in clustering techniques for complex data due to their superior feature extraction capabilities compared to traditional shallow methods. Generally, a feature extracting encoder such as deep neural networks is first applied to map the input data to an embedding space, then traditional clustering techniques such as $k$-means are applied to the embeddings to facilitate the downstream clustering tasks (Huang et al., 2014; Song et al., 2013). There are also several recent works (Xie et al., 2016; Yang et al., 2017; 2019; 2016; Li et al., 2021) that integrate the feature learning process and clustering into an end-to-end framework, which yield high performance for large-scale datasets.

While existing deep approaches have achieved notable success on clustering, they primarily focus on training a model to obtain optimal clustering performance on the data from a given domain. When data from a new domain is present, an interesting question is can we leverage the acquired knowledge from the learned model on trained domains to guide the clustering process in new domains. Unfortunately, existing deep clustering models can be hardly transferred from one domain to another. This limitation arises primarily from the fixed centroid-based learning approach employed by these methods. As illustrated in Figure 1, discrepancies often exist between the distributions of the source and target domains. Consequently, the learned fixed centroids may no longer be suitable for the target domain, leading to suboptimal clustering results. However, the process of training a new model from scratch for each domain incurs a substantial computational burden. More importantly, the acquired knowledge pertaining to the intra- and inter-clusters structure and patterns remains underutilized, impeding its potential to guide the clustering process on new data from similar domains. These limitations significantly hinder the practicability of deep clustering methods.

To address these limitations, there is a need for transferable deep clustering models that can leverage acquired knowledge from trained domains to guide clustering in new domains. By transferring the underlying principles of clustering on trained source domains, the model could learn how to cluster better and adapt such knowledge to clustering new data in the target domains. Unfortunately, there exists no trivial way to directly generalize existing deep clustering methods due to several

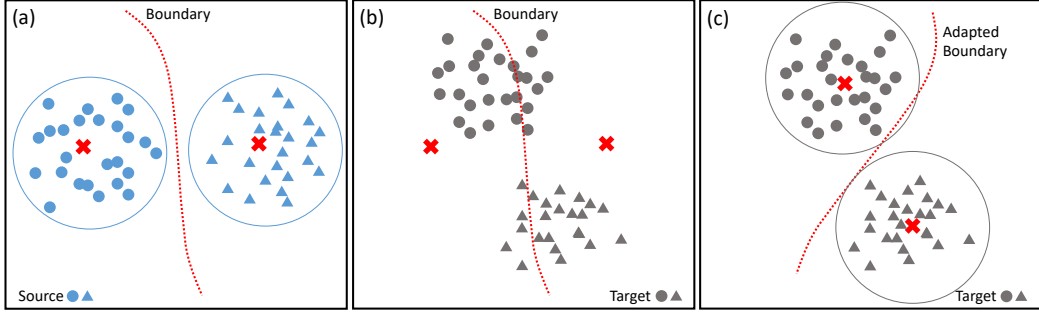

Figure 1: Problem of interest. (a) The cluster centroids learned from source domain can perfectly cluster source data samples; (b) The fixed centroids are not reliable to cluster target samples due to the distribution shift between source samples and target samples; (c) The cluster centroids are adapted to optimal position to better cluster the target samples. *Best viewed in color.*

major challenges: (1) **Difficulty in unsupervisely learning the shared knowledge among different domains.** In clustering scenarios, where labeled data is unavailable, extracting meaningful and transferable knowledge that capture the commonalities of underlying cluster structures across domains is challenging. (2) **Difficulty in ensuring the learned knowledge can be adapted and customized to target domains.** As shown in Figure 1(b), the distribution discrepancies between source and target domains can significantly harm the clustering performance of existing deep clustering models. Adapting the shared knowledge to new domains remains a challenging task in order to mitigate the negative impact of these distribution discrepancies. (3) **Difficulty in theoretically ensuring a stable learning process of clustering module.** Unlike supervised learning tasks, clustering models lack labeled data to provide guidance during training, making it even more crucial to establish theoretical guarantees for stability. Addressing this challenge requires developing theoretical frameworks that can provide insights into the stability and convergence properties of clustering algorithms.

In order to address the above metioned challenges, in this paper we propose a novel method named **T**ransferable **D**eep **C**lustering **M**odel (TDCM). To address the first challenge, we introduce an end-to-end learning framework that can jointly optimize the feature extraction encoder and a learnable clustering module. This framework aims to leverage the learned model parameter to capture the shared intra-cluster and inter-cluster structure derived from trained cluster patterns. Therefore, the shared knowledge can be effectively transferred to unseen data from new domains. To solve the second challenge, in stead of optimizing a fixed set of centroids, a novel learnable attention-based module is proposed for the clustering process to automatically adapt centroids to the new domains, as illustrated in the Figure 1(c). Therefore, the learned clustering model is not limited to the trained source domains and can be easily generalized to other domains. Specifically, this module enables the updating of centroids through a cluster-driven bi-partite attention block, allowing the model to be aware of the similarity relationships among data samples and capture the underlying structures and patterns. Furthermore, we provide theoretical evidence to demonstrate the strong expressive power of the proposed attention-based module in representing the relationships among data samples. Our theoretical analysis reveals that traditional centroid-based clustering models like $k$-means or GMM can be considered as special cases of our model. This theoretical proof highlights the enhanced capabilities of our approach compared to traditional clustering methods, emphasizing its potential for mining complex cluster patterns from data. Finally, we demonstrate the effectiveness of our proposed framework on both synthetic and real-world datasets. The experimental results show that our method can achieve strongly competitive clustering performance on unseen data by a single forward pass.

## 2 RELATED WORKS

**Deep clustering models.** Existing deep clustering methods can be classified into two main categories: separately and jointly optimization. The separately optimization methods typically first train a feature extractor by self-supervised task such as deep autoencoder models, then traditional clustering methods such as $k$-means (Huang et al., 2014), GMM (Yang et al., 2019) or spectral clustering (Affeldt et al., 2020) are applied to obtain the clustering results. There are also some works (Rodriguez & Laio, 2014) using density-based clustering algorithm such as DBSCAN (Ren et al., 2020) to avoid an explicit

choice of number of centroids. However, the separately methods require a two-step optimization which lack the ability to train the model in an end-to-end manner to learning representation that is more suitable for clustering. On the contrary, the jointly methods are becoming more popular in the era of deep learning. One prominent approach is the Deep Embedded Clustering (DEC) model (Xie et al., 2016), which leverages an autoencoder network to map data to a lower-dimensional representations and then optimize the clustering loss KL-divergence between the soft assignments of data to centroids and an adjusted target distribution with concentrated cluster assignments. Deep clustering model (DCN) (Yang et al., 2017) jointly optmize the dimensionality reduction and $k$-means clustering objective functions via learning a deep autoencoder and a set of $k$-means centroids in the embedding space. JULE (Yang et al., 2016) formulates the joint learning in a recurrent framework, which incorporates agglomerative clustering technique as a forward pass in neural networks. More recently, some works (Li et al., 2021; Yaling Tao, 2021; Niu et al., 2022) also propose to use contrastive learning by data augmentation techniques to obtain more discriminative representations for downstream clustering tasks.

However, most existing deep clustering methods focus on optimizing a fixed set of centroids, which limits their transferability as they struggle to handle distribution drift between different source and target domains. In contrast, our proposed model takes a different approach by adapting the centroids to learned latent embeddings, allowing it to be aware of distribution drift between domains and enhance its transferability.

**Attention models.** Attention models (Bahdanau et al., 2014; Vaswani et al., 2017; Han et al., 2022) have gained significant attention in the field of deep learning, revolutionizing various tasks across natural language processing, computer vision, and sequence modeling. These works collectively demonstrate the versatility and effectiveness of attention models in capturing informative relationships between data samples.

**Deep metric learning.** Our method is also related to deep metric learning methods that aim to learn representations from high-dimensional data in such a way that the similarity or dissimilarity between samples can be accurately measured. One prominent approach is the Contrastive Loss (Hadsell et al., 2006), which encourages similar samples to have smaller distances in the embedding space. Siamese networks (Chopra et al., 2005) learns embeddings by comparing pairs of samples and optimizing the contrastive loss. More recently, the Angular Loss (Wang et al., 2017) incorporates angular margins to enhance the discriminative power of the learned embeddings. Proxy-NCA (Movshovitz-Attias et al., 2017) employs proxy vectors to approximate the intra-class variations, enabling large-scale metric learning.

**Connection with Unsupervised Domain Adaption (UDA) methods.** While both our work and existing Unsupervised Domain Adaptation (UDA) methods (Ganin & Lempitsky, 2015; Long et al., 2016; Liang et al., 2020) involve transferring models from source domains to target domains, the primary goal of our paper differs significantly from UDA tasks. UDA methods assume the presence of labeled data in the source domains, allowing the model to be trained in a supervised manner. In contrast, our paper focuses on a scenario where no labels are available in the source domain, necessitating the use of unsupervised learning techniques. This key distinction highlights the unique challenges and approaches we address in our research.

## 3 PRELIMINARIES

In this section, we first formally define the problem formulation of transferable clustering task and then present the key challenges involved in designing an effective transferable deep clustering model.

In our study, we focus on a collection of datasets denoted as $\mathcal{D} = \{D_1, D_2, \ldots, D_m\}$. Each dataset $D_j$ is sampled from a joint probability distribution $p(\mathcal{D})$. Within each sampled dataset $D_j$, we have a set of high-dimensional feature vectors denoted as $D_j = \{\mathbf{x}_i^j\}_{i=1}^{N_j}$, where $\mathbf{x}_i^j$ represents the feature vector for the $i$-th sample. Our objective is to learn shared knowledge in clustering from a subset of datasets, referred to as the training set $D_s$ (source), and utilize this acquired knowledge to predict the clustering patterns on newly sampled unseen datasets, serving as the test set $D_t$ (target).

To achieve this, we aim to learn a clustering model denoted as $f$, trained on the source datasets $D_s$. The model $f$ partitions each source dataset $\{\mathbf{x}_i^s\}_{i=1}^{N_s}$ into $K$ clusters in an unsupervised manner, where $K$ is the desired number of clusters. Our goal is to maximize the intra-cluster similarities

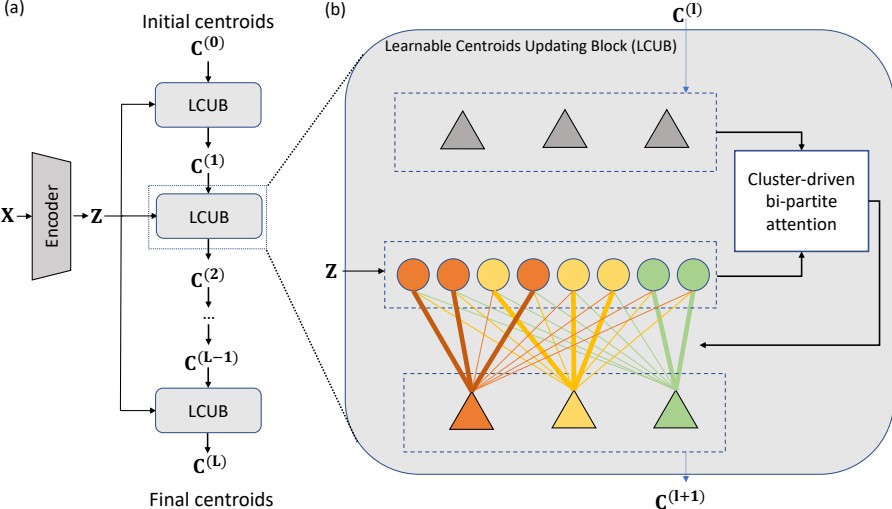

Figure 2: (a) The overall framework of the proposed approach. An encoder is first applied to extract latent embeddings $\mathbf{Z}$ from input samples $\mathbf{X}$. Then the initial centroids will forward pass a series of Learnable Centroids Updating Block (LCUB) to learn the underlying similarities with centroids to reveal the cluster patterns; (b) The detailed architecture of LCUB. The current cetroids $\mathbf{C}^{(l)}$ and latent sample embeddings $\mathbf{Z}$ are forwarded to form a bi-partile graph to calculate the assignment weights by pairwise attention scores, then the centroids are updated by the computed assignment weights.

and minimize the inter-cluster similarities by learning the clustering rule from the training datasets. Subsequently, we evaluate the clustering performance of the learned function $f$ on the test target sets $D_t$. By leveraging the knowledge acquired during training, we aim to accurately predict the cluster patterns in the test datasets.

## 4 METHODOLOGY

To address the aforementioned challenges, we propose a novel method named **T**ransferable **D**eep **C**lustering **M**odel (TDCM). To ensure the shared clustering knowledge among domains can be learned unsupervisely, we propose an end-to-end learning framework that jointly optimizes the feature extraction encoder and a learnable clustering module, as depicted in Figure 2(a). The framework aims to utilize the learned model parameters to capture the shared intra-cluster and inter-cluster structure derived from trained cluster patterns. Consequently, this enables effective transfer of the shared knowledge to unseen data from new domains. To adjust the learned knowledge to the target domains, in stead of optimizing a fixed set of centroids, a novel learnable attention-based module is proposed to automatically adapt centroids to the new domains, as shown in Figure 2(b). Therefore, the learned clustering model is not restricted to the trained source domains and can be easily generalized to other domains. Specifically, this module integrates a cluster-driven bi-partite attention block to update centroids, considering the similarity relationships among data samples and capturing underlying structures and patterns. Furthermore, we provide theoretical evidence to demonstrate the strong expressive power of the proposed attention-based module in representing the relationships among data samples. Our theoretical analysis reveals that traditional centroid-based clustering models like $k$-means or GMM can be considered as special cases of our model. This theoretical proof highlights the enhanced capabilities of our approach compared to traditional clustering methods, emphasizing its potential for mining complex cluster patterns from data.

### 4.1 TRANSFERRABLE CLUSTER CENTROIDS LEARNING FRAMEWORK

As previously discussed, existing deep clustering models typically treat centroids as fixed learnable parameters, which limits their ability to generalize effectively to unseen data. To address this limitation, we propose a novel clustering framework that can dynamically adjust the centroids based on the extracted sample embeddings. Consequently, the centroids are dynamically adapted based on the distribution of sample embeddings, endowing the model with the capability to effectively transfer to new domains. As depicted in Figure 2(a), an encoder $g$ is first utilized to extract latent embeddings $\mathbf{Z} = g_\phi(\mathbf{X}; \phi)$. Then the adaption process involves forward pass on a series of centroids

updating blocks: $\{\mathbf{c}_j^{(0)}\}_{j=1}^K \rightarrow \{\mathbf{c}_j^{(1)}\}_{j=1}^K \rightarrow \ldots \{\mathbf{c}_j^{(L)}\}_{j=1}^K$, where each block consists of two steps: assignment and update. In the assignment step of the $l$-th ($l \in [0, L]$) block, we compute the probability $\delta_{ij}$ that assigns the data sample $\mathbf{z}_i$ to the current cluster centroid $\mathbf{c}_j^{(l)}$ using a score function $\ell(\mathbf{z}_i, \mathbf{c}_j^{(l)})$, which captures the underlying similarity relationships among samples. Subsequently, we update the cluster centroids based on the assigned data points. The updating process can be mathematically formalized as:

$$
\begin{aligned}
\mathbf{c}_j^{(l+1)} &= \frac{1}{\sum_i^N \sum_j^K \delta_{ij}^{(l+1)}} \sum_{i=1}^N \delta_{ij}^{(l+1)} \mathbf{z}_i, \\
\delta_{ij}^{(l+1)} &= \frac{\exp\left(\ell(\mathbf{z}_i, \mathbf{c}_j^{(l)})/\tau\right)}{\sum_{j=1}^K \exp\left(\ell(\mathbf{z}_i, \mathbf{c}_j^{(l)})/\tau\right)},
\end{aligned}
\tag{1}
$$

where $\tau$ denotes the temperature hyper-parameter.

## 4.2 LEARNABLE CENTROIDS UPDATING MODULE

Given the overall updating procedure described earlier, a key consideration is the choice of the score function $\ell(\mathbf{z}_i, \mathbf{c}_j)$ to capture the similarity relationship between samples and centroids, thereby capturing the underlying cluster structure. Traditionally, a common approach is to use handcrafted score functions like the Euclidean distance $\ell(\mathbf{z}_i, \mathbf{c}_j) = \|\mathbf{z}_i - \mathbf{c}_j\|_2$. However, designing a specific score function requires domain knowledge and lacks generalizability across different domains.

To address this issue, we propose a learnable score function $\ell(\mathbf{z}_i, \mathbf{c}_j; \mathbf{W})$ by introducing learnable weights $\mathbf{W}$ that automatically capture the relational metrics between samples in a data-driven manner. Notably, the formulation in Equation 1 resembles a bi-partite graph structure of centroids and samples, which is illustrated in Figure 2(b). An attention-like mechanism, which selectively allocates resources based on the relevance of information, can be constructed based on the bi-partite structure. Since the goal of updating centroids is to gradually push centroids to represent a group of similar samples, ideally the score function $\ell(\mathbf{z}_i, \mathbf{c}_j; \mathbf{W})$ should achieve its maximum value when $\mathbf{z}_i = \mathbf{c}_j$. However, a common design of attention mechanism can not guarantee this property due to the arbitrary choice of learnable parameters $\mathbf{W}$ (see our proof in Appendix Theorem A.1).

To solve this issue from a theoretical perspective, we propose a novel clustering-driven bi-partite attention module with appropriate constraints on the parameters of learnable matrices. Specifically, the score function is designed as $\ell(\mathbf{z}_i, \mathbf{c}_j; \mathbf{W}_Q, \mathbf{W}_K) = -\sigma(\mathbf{W}_Q(\mathbf{z}_i - \mathbf{c}_j^{(l)}) \cdot \mathbf{W}_K(\mathbf{z}_i - \mathbf{c}_j^{(l)}))/\tau$ with two learnable weight matrices and we rewrite the Equation 1 as:

$$
\begin{aligned}
\delta_{ij}^{(l+1)} &= \frac{\exp\left(-\sigma(\mathbf{W}_Q(\mathbf{z}_i - \mathbf{c}_j^{(l)}) \cdot \mathbf{W}_K(\mathbf{z}_i - \mathbf{c}_j^{(l)}))/\tau\right)}{\sum_{j=1}^K \exp\left(-\sigma(\mathbf{W}_Q(\mathbf{z}_i - \mathbf{c}_j^{(l)}) \cdot \mathbf{W}_K(\mathbf{z}_i - \mathbf{c}_j^{(l)}))/\tau\right)}, \\
\mathbf{W}_Q &= \mathbf{W}_Q^{\mathsf{T}}, \mathbf{W}_K = \mathbf{W}_K^{\mathsf{T}},
\end{aligned}
\tag{2}
$$

where $\mathbf{W}_Q$ and $\mathbf{W}_K$ are two learnable real-symmetric matrices and $\sigma$ is a continuous non-decreasing nonlinear activation function (e.g. ReLU (Nair & Hinton, 2010) or LeakyReLU (Maas et al., 2013)).

**Theorem 4.1.** *The score function $\ell(\mathbf{z}_i, \mathbf{c}_j; \mathbf{W}_Q, \mathbf{W}_K) = -\sigma(\mathbf{W}_Q(\mathbf{z}_i - \mathbf{c}_j^{(l)}) \cdot \mathbf{W}_K(\mathbf{z}_i - \mathbf{c}_j^{(l)}))/\tau$ defined in Equation 2 can guarantee that $\forall \mathbf{z}_i \in \mathbb{R}^b$, we have $\ell(\mathbf{z}_i, \mathbf{c}_j) \leq \ell(\mathbf{c}_j, \mathbf{c}_j)$.*

*Proof.* We first define $\mathbf{p} = \mathbf{z}_i - \mathbf{c}_j^{(l)}$ and rewrite the score function as $\ell = -\sigma(\mathbf{W}_Q \mathbf{p} \cdot \mathbf{W}_K \mathbf{p})/\tau$. We rewrite the inner product part as

$$
\mathbf{W}_Q \mathbf{p} \cdot \mathbf{W}_K \mathbf{p} = (\mathbf{p} \mathbf{W}_Q)^{\mathsf{T}} \cdot \mathbf{W}_K \mathbf{p} = \mathbf{p}^{\mathsf{T}}(\mathbf{W}_Q^{\mathsf{T}} \mathbf{W}_K) \mathbf{p}.
$$

Since $\mathbf{W}_Q$ and $\mathbf{W}_Q$ are two real-symmetric matrices, $\mathbf{W}_Q^{\mathsf{T}} \mathbf{W}_K$ is a positive-definite matrix. For any nonzero real vector $\mathbf{p}$, we have $\mathbf{p}^{\mathsf{T}}(\mathbf{W}_Q^{\mathsf{T}} \mathbf{W}_K) \mathbf{p} > 0$. In addition, due to the property of continuous and non-decreasing, the nonlinear activation function would not change the ordering of values. Therefore, for all $\mathbf{z}_i \in \mathbb{R}^b$, we have $\ell(\mathbf{z}_i, \mathbf{c}_j) \leq \ell(\mathbf{c}_j, \mathbf{c}_j)$. $\square$

In addition to theoretical property that our centroids updating module can group similar samples within same clusters, we further prove that our defined score function in Equation 2 can theoretically have stronger expressive power in representing the similarity relationship between data samples than traditional clustering technique such as $k$-means or GMM by the following theorems:

**Theorem 4.2.** *The score function of $k$-means and GMM models are special cases of our defined score function $\ell(\mathbf{z}_i, \mathbf{c}_j; \mathbf{W}_Q, \mathbf{W}_K)$ in Equation 2.*

The proof for $k$-means algorithm is straightforward given here and the proof for GMM models can be found in the Appendix.

*Proof.* By setting the nonlinear function $\sigma$ as identity function and both $\mathbf{W_Q}$ and $\mathbf{W_K}$ as identity matrix $\mathbf{I}$, we can rewrite the score function as $\ell(\mathbf{z}_i, \mathbf{c}_j) = -\|\mathbf{z}_i - \mathbf{c}_j\|_2^2/\tau$, which is the negative squared Euclidean distance. Then the model is equalize to a soft $k$-means centroids updating step. By setting $\tau \to 0^+$, the process converges to the traditional $k$-means algorithm. $\qquad\square$

### 4.3 Unsupervised learning objective function

In order to optimize the parameters of the proposed model, the overall objective function of our framework can be written as:

$$\min_{g_\phi, \mathbf{W}_Q, \mathbf{W}_K} \mathcal{L}_{\text{clustering}} + \beta \mathcal{L}_{\text{entropy}}. \tag{3}$$

Here the first term $\mathcal{L}_{\text{clustering}}$ is aimed at maximizing the similarity scores within clusters:

$$\mathcal{L}_{\text{clustering}} = -\sum_l^L \alpha^{(l)} \sum_i^N \sum_j^K \delta_{ij}^{(l)} \ell(g_\phi(\mathbf{x}_i), \mathbf{c}_j^{(l)}; \mathbf{W}_Q, \mathbf{W}_K),$$
$$s.t. \mathbf{W}_Q \mathbf{W}_Q^\intercal = \mathbf{I}, \mathbf{W}_K \mathbf{W}_K^\intercal = \mathbf{I} \tag{4}$$

where $\alpha^{(l)}$ are hyperparameters to tune the balance between blocks and orthogonal constraints are incorporated to prevent the trivial solution of scale changes in the embeddings. We can treat the constraints as a Lagrange multiplier and solve an equivalent problem by substituting the constraint to a regularization term.

Besides the clustering loss term, the entropy loss term is aimed at avoiding the trivial solution of assigning all samples to one single cluster:

$$\mathcal{L}_{\text{entropy}} = -\sum_l^L \alpha^{(l)} \sum_j^K \pi_j^{(l)} \log \pi_j^{(l)},$$
$$\pi_j^{(l)} = \sum_i \delta_{ij}^{(l)} = \sum_i \frac{\exp\left(-\sigma(\mathbf{W}_Q(\mathbf{z}_i - \mathbf{c}_j^{(l-1)}) \cdot \mathbf{W}_K(\mathbf{z}_i - \mathbf{c}_j^{(l-1)}))/\tau\right)}{\sum_{j=1}^K \exp\left(-\sigma(\mathbf{W}_Q(\mathbf{z}_i - \mathbf{c}_j^{(l-1)}) \cdot \mathbf{W}_K(\mathbf{z}_i - \mathbf{c}_j^{(l-1)}))/\tau\right)}, \tag{5}$$

where $\pi_j^{(l)}$ reflects the size of each clusters.

**Initialization of centroids.** Many previous studies use the centroids provided by traditional clusteing methods such as $k$-means on the latent embeddings as the initialization of centroids. However, these methods usually requires to load all data samples into the memory, which can be hardly generalize to a mini-batch version due to the permutation invariance of cluster centroids. To solve this issue, we propose to initialize the centroids $\{\mathbf{c}_j^{(0)}\}_{j=1}^K$ before blocks as a set of orthogonal vectors in the embedding space, e.g. identity matrix $\mathbf{I}$.

### 4.4 Complexity analysis

Here we present the complexity analysis of our proposed dynamic centroids update module. In each block, we need to compute the pair-wise scores between centroids and data samples in Equation 2. Assuming the embedding space dimension is denoted as $b$, the time complexity to calculate the score functions in one block is $O(NKb^2)$. Consequently, performing $L$ blocks would entail a time complexity of $O(LNKb^2)$, where $N$ represents the number of samples and $K$ denotes the number of centroids. It is important to note that our framework naturally supports a mini-batch version, which significantly enhances the scalability of the model and improves its efficiency.

## 5 EXPERIMENTS

In this section, the experimental settings are introduced first in Section 5.1, then the performance of the proposed method on synthetic datasets are presented in Section 5.2. We further present the effectiveness test on our method against distributional shift between domains on real-world datasets in Section 5.3. In addition, we verify the effectiveness of framework components through ablation studies in Section 5.4. Due to space limit, we also include additional experiments on the datasets with more categories in Appendix B.1, and measure the parameter sensitivity in Appendix B.2

### 5.1 EXPERIMENTAL SETTINGS

**Synthetic datasets.**  In order to assess the generalization capability of our proposed method towards unseen domain data, we conduct an evaluation using synthetic datasets. A source domain is first generated by sampling $K$ equal-sized data clusters. The data features are sampled from multi-Gaussian distributions with randomized centers and covariance matrices, which is similar to previous works (Karimi et al., 2018; Chen et al., 2018). Subsequently, a corresponding target domain is created by randomly perturbing the centers of the source domain clusters. This ensures the presence of distributional drift between the train and test set data. To provide comprehensive results, we vary the value of $K$ and generate 10 distinct datasets for each value of $K$. We train the clustering model on source domain and test on the target domain. Our experimental results are reported as an average of 5 runs on each dataset, with different random seeds employed to ensure robustness.

**Real-world datasets.**  To further evaluate the generalization capability of our proposed method under real-world senarios, commonly used real-world benchmark datasets are included. (1) **Digits** which includes MNIST and USPS, is a standard digit recognition benchmark that commonly used by previous studies (Xie et al., 2016; Yang et al., 2017; Long et al., 2018; Liang et al., 2020). Follow previous works (Long et al., 2018; Liang et al., 2020), we train the model on the source domain training set and test the model on the target domain test set. All input images are resized to $32 \times 32$. (2) **CIFAR-10** (Krizhevsky et al., 2009) is commonly used image benchmark datasets in evaluating deep clustering models. We treat the training set as source domain and test set as target domain. We introduce CenterCrop to the test set to create distribution drift.

**Comparison methods.**  We evaluate the proposed method on both synthetic and real-world benchmark datasets and compare it with both traditional clustering and state-of-the-art deep clustering techniques such as $k$-means, GMM, DAE (Vincent et al., 2010), DAEGMM (Wang & Jiang, 2021), DEC (Xie et al., 2016), DCN (Yang et al., 2017), JULE (Yang et al., 2016), CC (Li et al., 2021) and IDFD (Yaling Tao, 2021).

**Evaluation metrics.**  In our evaluation of clustering performance, we employ widely recognized metrics, namely normalized mutual information (NMI) (Cai et al., 2010), adjusted rand index (ARI) (Yeung & Ruzzo, 2001), and clustering accruracy (ACC) (Cai et al., 2010). By combining NMI, ARI, and ACC, we can comprehensively demonstrate the efficacy of our clustering results.

**Implementation details.**  Our proposed model serves as a general framework, allowing for the integration of various commonly used deep representation learning techniques as the encoder part. To ensure a fair comparison with previous works, we enforce the use of the same encoder for feature extraction across all models. Specifically, for synthetic data, we utilize a three-layer multilayer perceptron (MLP) as the encoder. For the Digits dataset, we employ the classical LeNet-5 network (LeCun et al., 1998) as the encoder. Furthermore, for the CIFAR-10 datasets, we utilize the ResNet-18 network (He et al., 2016) as the encoder. We use $L = 4$ layers of blocks to update the synthetic datasets and $L = 5$ for the real-world datasets. The temperature $\tau$ is set as $1.0$ throughout the whole experiments. We use an linearly increasing series of values for the weights $\alpha$ for penalizing each block in loss function, where the final layer has the largest weight. We train the whole network through back-propagation and utilize Adam (Kingma & Ba, 2014) as the optimizer. The initial learning rate is set as $5e^{-3}$ for the synthetic datasets and $5e^{-4}$ for the real-world datasets, and the weight decay rate is set as $5e^{-4}$. The total number of training epochs is $500$ for the synthetic datasets and $2,000$ for the real-world datasets. The batch size is set as $256$ for synthetic and DIGITS datasets, and $128$ for CIFAR-10 dataset. Data augmentation techniques are added like previous papers (Li et al., 2021; Yaling Tao, 2021) for the purpose of training discriminative representations for all the

Table 1: Clustering performance on the synthetic datasets with varying number of clusters $K$. The models are trained on source domain and tested on target domain. We denote the performance drop from training set to test set as 'diff', where smaller values indicate better generalization ability. The best generalization performances are highlighted in bold.

| model | | K=2 | | | K=3 | | | K=5 | | | K=10 | | |
|---|---|---|---|---|---|---|---|---|---|---|---|---|---|
| | | NMI | ARI | ACC | NMI | ARI | ACC | NMI | ARI | ACC | NMI | ARI | ACC |
| | source | 0.995 | 0.998 | 0.999 | 0.951 | 0.930 | 0.940 | 0.898 | 0.845 | 0.855 | 0.924 | 0.850 | 0.845 |
| $k$-means | target | 0.622 | 0.596 | 0.828 | 0.883 | 0.846 | 0.916 | 0.750 | 0.653 | 0.745 | 0.782 | 0.643 | 0.731 |
| | diff | 0.373 | 0.402 | 0.171 | 0.068 | 0.084 | 0.024 | 0.148 | 0.192 | 0.110 | 0.142 | 0.207 | 0.114 |
| | source | 0.995 | 0.998 | 0.999 | 0.991 | 0.995 | 0.998 | 0.934 | 0.919 | 0.936 | 0.953 | 0.919 | 0.924 |
| GMM | target | 0.622 | 0.597 | 0.824 | 0.902 | 0.877 | 0.948 | 0.756 | 0.674 | 0.775 | 0.788 | 0.661 | 0.755 |
| | diff | 0.373 | 0.401 | 0.175 | 0.089 | 0.118 | 0.050 | 0.178 | 0.245 | 0.161 | 0.165 | 0.258 | 0.169 |
| | source | 0.995 | 0.998 | 0.999 | 0.949 | 0.928 | 0.939 | 0.899 | 0.852 | 0.861 | 0.919 | 0.855 | 0.854 |
| AE | target | 0.532 | 0.516 | 0.778 | 0.754 | 0.746 | 0.824 | 0.642 | 0.635 | 0.701 | 0.702 | 0.613 | 0.712 |
| | diff | 0.463 | 0.482 | 0.221 | 0.195 | 0.182 | 0.115 | 0.257 | 0.217 | 0.160 | 0.217 | 0.242 | 0.142 |
| | source | 0.996 | 0.998 | 0.998 | 0.990 | 0.993 | 0.989 | 0.933 | 0.922 | 0.933 | 0.945 | 0.908 | 0.916 |
| DAEGMM | target | 0.522 | 0.507 | 0.713 | 0.842 | 0.827 | 0.878 | 0.696 | 0.704 | 0.734 | 0.690 | 0.610 | 0.687 |
| | diff | 0.474 | 0.491 | 0.285 | 0.148 | 0.166 | 0.111 | 0.237 | 0.218 | 0.199 | 0.255 | 0.298 | 0.229 |
| | source | 0.995 | 0.998 | 0.999 | 0.989 | 0.991 | 0.997 | 0.932 | 0.942 | 0.978 | 0.955 | 0.942 | 0.973 |
| DEC | target | 0.692 | 0.636 | 0.828 | 0.889 | 0.851 | 0.905 | 0.766 | 0.683 | 0.785 | 0.701 | 0.605 | 0.713 |
| | diff | 0.303 | 0.362 | 0.171 | 0.100 | 0.140 | 0.092 | 0.166 | 0.259 | 0.193 | 0.254 | 0.337 | 0.260 |
| | source | 0.994 | 0.997 | 0.999 | 0.991 | 0.991 | 0.997 | 0.937 | 0.950 | 0.982 | 0.963 | 0.954 | 0.978 |
| DCN | target | 0.654 | 0.498 | 0.719 | 0.703 | 0.608 | 0.795 | 0.643 | 0.698 | 0.742 | 0.689 | 0.599 | 0.753 |
| | diff | 0.340 | 0.499 | 0.280 | 0.288 | 0.383 | 0.202 | 0.294 | 0.252 | 0.240 | 0.274 | 0.355 | 0.225 |
| | source | 0.990 | 0.992 | 0.998 | 0.980 | 0.981 | 0.998 | 0.938 | 0.950 | 0.981 | 0.962 | 0.963 | 0.982 |
| CC | target | 0.578 | 0.555 | 0.694 | 0.821 | 0.802 | 0.854 | 0.623 | 0.634 | 0.701 | 0.694 | 0.645 | 0.721 |
| | diff | 0.412 | 0.437 | 0.304 | 0.159 | 0.179 | 0.144 | 0.315 | 0.316 | 0.280 | 0.268 | 0.318 | 0.261 |
| | source | 0.990 | 0.991 | 0.998 | 0.975 | 0.982 | 0.994 | 0.935 | 0.949 | 0.979 | 0.961 | 0.965 | 0.984 |
| TDCM | target | 0.989 | 0.995 | 0.999 | 0.953 | 0.957 | 0.984 | 0.901 | 0.896 | 0.951 | 0.885 | 0.863 | 0.925 |
| | diff | **0.001** | **-0.004** | **-0.001** | **0.022** | **0.025** | **0.010** | **0.034** | **0.053** | **0.028** | **0.076** | **0.102** | **0.059** |

image datasets. The experiments are carried out on NVIDIA A6000 GPUs, which takes around 30 gpu-hours to train the model on CIFAR-10 dataset.

## 5.2 SYNTHETIC DATA RESULTS

Table 1 presents the clustering performance on both the trained source domain and test target domain of synthetic datasets. The results show the remarkable effectiveness of our proposed TDCM framework in achieving superior generalization performance when transferring the trained model from source to target sets across all synthetic scenarios. Specifically, TDCM consistently outperforms all the comparison methods, exhibiting an average improvement of 0.215, 0.243, and 0.157 on NMI, ARI, and ACC metrics, respectively. Notably, the performance of the TDCM model on the test set exhibits only a marginal average decrease of 0.033, 0.044, and 0.024 on NMI, ARI, and ACC metrics, respectively, compared to the training set. These results provide strong evidence that our proposed method significantly enhances the transferability of the clustering model, demonstrating its superior performance and robustness. On the other hand, although the comparison methods can achieve competitive performance on the trained training set, their performance drops significantly when transfer from source to target domains, which proves that their fixed set of optimized centroids can not handle the distribution drift between domains.

## 5.3 REAL-WORLD DATA RESULTS

We report the clustering results of the real-world datasets in Table 2. The results demonstrate the strength of our proposed TDCM framework by consistently achieving the best performance when test on test sets across all datasets. Specifically, TDCM consistently outperforms all the comparison methods, exhibiting an average improvement of 0.206, 0.342, 0.439 on MNIST, USPS, and CIFAR-10 data test sets, respectively. Our results strongly demonstrate the enhanced transferability of our proposed method for the clustering model, highlighting its superior performance. It worth noting that the improvement of our model on CIFAR-10 dataset is more significant than the other two digits dataset. A possible reason is CIFAR-10 datasets are more complex than the other two datasets, which may prove that our model can handle complex data with high dimensional features.

## 5.4 ABLATION STUDIES

Here we investigate the impact of the proposed components of TDCM. We first consider variants of removing the real-symmetric constraints and orthogonal constraints in our model, named *variant-R*

Table 2: Clustering performance on the real-world datasets. The models are trained on source domain and tested on target domain. The best performances on test datasets are highlighted in bold.

| | model | k-means | | GMM | | AE | | DEC | | DCN | | JULE | | CC | | IDFD | | TDCM | |
|---|---|---|---|---|---|---|---|---|---|---|---|---|---|---|---|---|---|---|---|
| | | source | target | source | target | source | target | source | target | source | target | source | target | source | target | source | target | source | target |
| MNIST | NMI | 0.503 | 0.432 | 0.466 | 0.387 | 0.808 | 0.774 | 0.804 | 0.789 | 0.811 | 0.779 | 0.913 | 0.874 | 0.932 | 0.881 | 0.921 | 0.898 | 0.925 | **0.933** |
| | ARI | 0.476 | 0.382 | 0.398 | 0.343 | 0.765 | 0.698 | 0.789 | 0.764 | 0.768 | 0.730 | 0.874 | 0.849 | 0.873 | 0.864 | 0.88 | 0.841 | 0.873 | **0.886** |
| | ACC | 0.535 | 0.501 | 0.465 | 0.431 | 0.797 | 0.754 | 0.849 | 0.822 | 0.831 | 0.818 | 0.963 | 0.907 | 0.945 | 0.895 | 0.951 | 0.931 | 0.938 | **0.958** |
| USPS | NMI | 0.607 | 0.496 | 0.633 | 0.451 | 0.593 | 0.446 | 0.582 | 0.387 | 0.856 | 0.495 | 0.881 | 0.872 | 0.902 | 0.652 | 0.913 | 0.754 | 0.905 | **0.911** |
| | ARI | 0.597 | 0.503 | 0.623 | 0.402 | 0.549 | 0.389 | 0.565 | 0.234 | 0.834 | 0.324 | 0.858 | 0.840 | 0.889 | 0.613 | 0.901 | 0.746 | 0.885 | **0.879** |
| | ACC | 0.611 | 0.587 | 0.654 | 0.558 | 0.610 | 0.537 | 0.605 | 0.451 | 0.869 | 0.598 | 0.913 | 0.783 | 0.908 | 0.705 | 0.937 | 0.803 | 0.925 | **0.910** |
| CIFAR-10 | NMI | 0.087 | 0.076 | 0.095 | 0.084 | 0.239 | 0.143 | 0.257 | 0.143 | 0.243 | 0.124 | 0.192 | 0.108 | 0.705 | 0.548 | 0.711 | 0.578 | 0.687 | **0.664** |
| | ARI | 0.049 | 0.033 | 0.062 | 0.049 | 0.169 | 0.114 | 0.161 | 0.094 | 0.143 | 0.079 | 0.138 | 0.087 | 0.637 | 0.421 | 0.663 | 0.467 | 0.642 | **0.617** |
| | ACC | 0.229 | 0.178 | 0.253 | 0.230 | 0.314 | 0.251 | 0.301 | 0.231 | 0.275 | 0.194 | 0.272 | 0.201 | 0.790 | 0.620 | 0.815 | 0.639 | 0.795 | **0.773** |

Table 3: Ablation studies. Values in parentheses indicate standard deviation.

| | | Synthetic K=2 | | | Synthetic K=5 | | | CIFAR-10 | | |
|---|---|---|---|---|---|---|---|---|---|---|
| | | NMI | ARI | ACC | NMI | ARI | ACC | NMI | ARI | ACC |
| Full model | source | 0.990(0.018) | 0.991(0.013) | 0.998(0.001) | 0.935(0.001) | 0.949(0.001) | 0.979(0.001) | 0.687(0.045) | 0.642(0.041) | 0.795(0.032) |
| | target | 0.989(0.017) | 0.995(0.007) | 0.999(0.002) | 0.901(0.035) | 0.896(0.026) | 0.951(0.013) | 0.664(0.061) | 0.617(0.057) | 0.773(0.043) |
| variant-R | source | 0.992(0.016) | 0.991(0.015) | 0.997(0.001) | 0.926(0.014) | 0.935(0.015) | 0.978(0.008) | 0.276(0.172) | 0.240(0.201) | 0.359(0.135) |
| | target | 0.678(0.234) | 0.541(0.301) | 0.698(0.197) | 0.754(0.123) | 0.721(0.141) | 0.805(0.067) | 0.178(0.087) | 0.159(0.102) | 0.246(0.155) |
| variant-O | source | 0.987(0.023) | 0.976(0.025) | 0.991(0.009) | 0.928(0.015) | 0.935(0.017) | 0.970(0.008) | 0.236(0.092) | 0.205(0.071) | 0.272(0.045) |
| | target | 0.878(0.064) | 0.842(0.070) | 0.898(0.045) | 0.851(0.073) | 0.821(0.091) | 0.902(0.027) | 0.148(0.075) | 0.139(0.042) | 0.186(0.055) |
| variant-E | source | 0.993(0.012) | 0.991(0.013) | 0.998(0.002) | 0.935(0.001) | 0.949(0.001) | 0.975(0.004) | 0.547(0.120) | 0.492(0.141) | 0.655(0.102) |
| | target | 0.969(0.037) | 0.955(0.039) | 0.981(0.012) | 0.891(0.025) | 0.886(0.031) | 0.936(0.020) | 0.564(0.161) | 0.417(0.134) | 0.597(0.143) |

and *variant-O*. In addition, we also remove the entropy loss in our overall loss function, named *variant-E*. We present the results on two synthetic datasets ($K = 2, 5$) and CIFAR-10 real-world dataset in Table 3, where we can observe a significant performance drop consistently for all variants. Especially, we observe that the standard deviation of all variants are larger than the full model, especially for the *variant-R* that removes real-symmetric constraint. Such behavior may demonstrate the importance of these proposed constraints in guaranteeing a stable training process, which is highly consistent with our theoretical analysis.

## 5.5 VISUALIZATION OF CENTROIDS UPDATING PROCESS

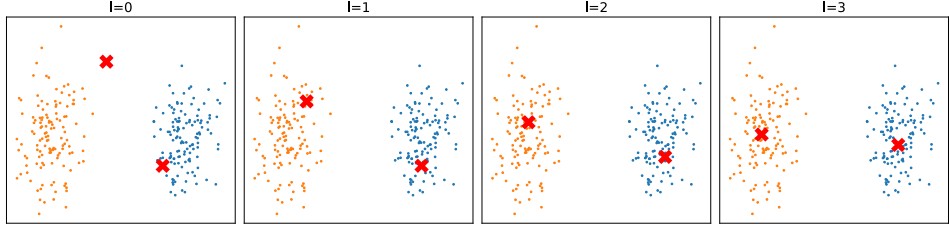

Figure 3: A visualization of centroids updating process with $L = 3$ blocks. Here orange and blue points denote two clusters and the red point denotes the centroids.

In order to illustrate the learned centroids updating behavior by our designed module, here we visualize the layer-wise updated centroids in $K = 2$ synthetic dataset test set in Figure 3. From the visualization we can observe that by forward passing the updating blocks, the centroids are adapted to a more clear cluster structure by the learned similarity metrics. It worth noting that the final adapted centroids are not necessarily at the 'center' of the clusters, which demonstrate that the designed module can automatically find the underlying similarity metric between samples.

## 6 CONCLUSIONS

In this study, we introduce a novel framework called **T**ransferable **D**eep **C**lustering **M**odel (TDCM) to tackle the challenge of limited generalization ability in previous end-to-end deep clustering techniques when faced with unseen domain data. In stead of optimizing a fixed set of centroids specific to the training source domain, our proposed TDCM employs an adapted centroids updating module, enabling automatic adaptation of centroids based on the input domain data. As a result, our framework exhibits enhanced generalization capabilities to handle unseen domain data. To capture the intrinsic structure and patterns of clusters, we propose an attention-based learnable module, which learns a data-driven score function for measuring the underlying similarity among samples. Theoretical analysis guarantees the effectiveness of our proposed module in extracting underlying similarity relationships, surpassing conventional clustering techniques such as $k$-means or Gaussian Mixture Models (GMM) in terms of expressiveness. Extensive experiments conducted on synthetic and real-world datasets validate the effectiveness of our proposed model in addressing distributional drift during the transfer of clustering knowledge from trained source domains to unseen target domains.

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
