# OpenReview forum: "Transferable Deep Clustering Model"
_ICLR.cc/2024/Conference — Submitted to ICLR 2024_

### Official Review · Reviewer_akM3 · 2023-10-13

**Soundness:** 2 fair
**Presentation:** 3 good
**Contribution:** 2 fair
**Rating:** 5
**Confidence:** 4

**Summary:**

The authors focused on the transferability of deep clustering models. They proposed a  model that can automatically adapt the cluster centroids according to the distribution of data samples, which is experimentally better than some existing methods on both synthetic data and real-world data. They also gave some theoretical results.

**Strengths:**

1. The paper is well organized and easy to follow
2. The topic is new, the transferability of deep clustering models haven't be addressed by previous methods
3. The learnable centroid update module is new
4. The improvement is significant on synthetic data

**Weaknesses:**

1. Although the transferability of deep clustering models haven't be addressed by previous methods, the motivation of this paper is not clear enough. Since clustering is an unsupervised task, why we just use the trained clustering model as the backbone and fine-tune on the target data.
2. For both synthetic dataset and real-world dataset, the cluster size is balanced and the number of clusters is fixed across source data and target data. The assumption is unreal in reality.
3. Several import baselines are missing, such as SCAN[1], GCC[2] and TCC[3]

[1]Van Gansbeke W, Vandenhende S, Georgoulis S, et al. Scan: Learning to classify images without labels[C]//European conference on computer vision. Cham: Springer International Publishing, 2020: 268-285.
[2]Zhong H, Wu J, Chen C, et al. Graph contrastive clustering[C]//Proceedings of the IEEE/CVF international conference on computer vision. 2021: 9224-9233.
[3]Shen Y, Shen Z, Wang M, et al. You never cluster alone[J]. Advances in Neural Information Processing Systems, 2021, 34: 27734-27746.

**Questions:**

1. Although the transferability of deep clustering models haven't be addressed by previous methods, the motivation of this paper is not clear enough. Since clustering is an unsupervised task, why we just use the trained clustering model as the backbone and fine-tune on the target data. Please discuss it.
2. For both synthetic dataset and real-world dataset, the cluster size is balanced and the number of clusters is fixed across source data and target data. The assumption is unreal in reality. Please discuss it.
3. Several import baselines are missing, such as SCAN[1], GCC[2] and TCC[3].

[1]Van Gansbeke W, Vandenhende S, Georgoulis S, et al. Scan: Learning to classify images without labels[C]//European conference on computer vision. Cham: Springer International Publishing, 2020: 268-285.
[2]Zhong H, Wu J, Chen C, et al. Graph contrastive clustering[C]//Proceedings of the IEEE/CVF international conference on computer vision. 2021: 9224-9233.
[3]Shen Y, Shen Z, Wang M, et al. You never cluster alone[J]. Advances in Neural Information Processing Systems, 2021, 34: 27734-27746.

---

### Official Review · Reviewer_qTtt · 2023-10-28

**Soundness:** 2 fair
**Presentation:** 2 fair
**Contribution:** 2 fair
**Rating:** 3
**Confidence:** 4

**Summary:**

This paper introduces a novel framework TDCM to tackle the challenge of limited generalization ability in previous end-to-end deep clustering techniques when faced with unseen domain data. The proposed model makes sense. However, many defects can be corrected.

**Strengths:**

1. The proposed model makes sense.

**Weaknesses:**

1. Is there any obvious advantage to using TDCM instead of using an unsupervised model on the target domain? Is it faster or does it have less cost?
2. How to decide the number of updating blocks L?
3. The experiment is too weak. 1) For the synthetic dataset, how about changing the number of centroids for the target domain? How about changing the size of each clustering? 2）Only two simple real-world datasets are used. A larger and more complicated dataset should be used. 3) For CIFAR-10, using only CenterCrop to create a target domain is too weak.
4. The last paragraph of Sec. Introduction and the first paragraph of Sec. The methodology is of highly repetitive content.

**Questions:**

1. Is there any obvious advantage to using TDCM instead of using an unsupervised model on the target domain? Is it faster or does it have less cost?
2. How to decide the number of updating blocks L?
3. The experiment is too weak. 1) For the synthetic dataset, how about changing the number of centroids for the target domain? How about changing the size of each clustering? 2）Only two simple real-world datasets are used. A larger and more complicated dataset should be used. 3) For CIFAR-10, using only CenterCrop to create a target domain is too weak.
4. The last paragraph of Sec. Introduction and the first paragraph of Sec. The methodology is of highly repetitive content.

---

### Official Review · Reviewer_ffVS · 2023-10-31

**Soundness:** 3 good
**Presentation:** 2 fair
**Contribution:** 2 fair
**Rating:** 6
**Confidence:** 3

**Summary:**

This paper propose an innovative approach to address the domain transfer problem in the field of deep clustering. This method primarily involves the introduction of an attention-based module, which automatically adjusts cluster centroids based on the distribution of data samples. In contrast to traditional fixed-centroid learning methods, this approach exhibits greater generality as it can adapt to the data distribution of new domains. Further experiments demonstrate a significant enhancement in the performance of the target domain and a reduction in computational costs.

**Strengths:**

1. This paper proposes a method to address the issue of domain transfer in the deep clustering and provides theoretical analysis.

2. The targeting problem is an inherent challenge of deep clustering. The paper provides an in-depth analysis of the domian shfit problem in Introduction section.

3. The paper has a well-organized structure, and the explanation of the algorithm design is relatively clear.

**Weaknesses:**

1. The method proposed may introduce an increased complexity to the model, potentially resulting in longer training times and higher computational demands.

2. The experiments were conducted on some small datasets. The limited size of the current dataset may hinder a comprehensive evaluation of the proposed method.

3. The methods compared in the experiments of this paper may not be the most relevant to the target problem. Therefore, the experiments may not prove that TDCM is SOTA.

4. This paper lacks an explanation of the method's effectiveness in addressing the target problem. For example, visualization of features in different domains.

5. The potential avenues for future research in this paper are not clearly delineated, leaving readers uncertain about the prospects of this work.

**Questions:**

1. In Eq.1, please explain the significance of the denominator in $\mathbf{c}_j^{(l+1)}

2. In the inner product section of Theorem 4.1, what are the respective dimensions of $\mathbf{p}$ and $\mathbf{W}_Q$ and why can they be multiplied like $\mathbf{p} \mathbf{W}_Q$?

3.Does the choice of the non-linear activation function σ in the formula affect the performance of the score function?

4.Are there any other transferable clustering models, and has this model been compared to them in terms of performance?

5.Has the model's performance been demonstrated on a subset of the ImageNet dataset?

6.Is there an open-source implementation of the experimental code or the model provided?

7. What range was the hyperparameter $\beta$ adjusted within during the experiments?

8. The proposed method claims to overcome the challenge of domain shift, why weren't some examples provided to demonstrate the superiority of TDCM?

---

### Official Review · Reviewer_UH8p · 2023-11-01

**Soundness:** 2 fair
**Presentation:** 2 fair
**Contribution:** 2 fair
**Rating:** 3
**Confidence:** 3

**Summary:**

In this paper, a transferrable clustering algorithm has been proposed, which is to improve the clustering performance on the target domain without any additional fine-tuning. To this end, the learnable cluster updating module is developed. It sequentially updates the centroids based on similarity, which are computed using learnable function, between centroids and instances in the embedding space. The proposed algorithm shows their performances on the synthetic and real world datasets.

**Strengths:**

1. The proposed algorithm aims to tackle a new clustering problem, which is to obtain good clustering results on the new domain without any fine-tuning.

2. It provides some theoretical analysis of the proposed algorithm and the corresponding proofs.

3. It shows the experimental results on various datasets, including synthetic and real-world datasets.

**Weaknesses:**

1. How the scores for the other methods are computed in Table 1(synthetic dataset experiment)? For example, are the centroids of $k$-means method re-computed on the target domain? I wonder if the scores are obtained by just using the nearest neighbor rule with the centroids obtained on the source domain. If so, I don't think it is a fair comparison because the proposed algorithm re-computed their centroids on the target centroids. The centroids of $k$-means also can be easily updated without fine-tuning or something.

2. I'm not sure that the experiment scenario is aligned with the objective of the proposed algorithm. It uses training set and test set as source and target domain, respectively. However, in general, they are from the same distribution. So, I think that the experimental results are not convincing enough to support the claims in the paper.

3. For the same reason I mentioned above (in 2), it would be much better to show the cross-domain adaptation results. For example, trained on MNIST and tested on DIGITS.

4. Thus, in overall, I think that the novelty of the proposed algorithm is not enough as well as the experimental results. The scoring function with the learnable matrix is not novel and centroid update module is also basically the same with the k-means except for the scoring function. Also, the experimental results are only provided on a few toy datasets. So, in my opinion, some more experimental results on more diverse datasets (not toy datasets) should be provided to validate the performance of the proposed algorithm.

5. I think the theorems are very straight forward in overall. So, of course, it can inspire some readers but I don't think they are meaningful enough to be considered as contribution of this paper.

**Questions:**

Please see the weakness section.

---

### Meta-Review · Area_Chair_4ucZ · 2023-12-06

**Metareview:**

This paper presents a approach to tackle the domain transfer problem for deep clustering. The proposed method introduces an attention-based module that dynamically adjusts cluster centroids based on the distribution of data samples. However, the experiments conducted on relatively small datasets pose limitations to a comprehensive evaluation of the proposed method due to the dataset's limited size. Furthermore, the methods compared in the experiments are not state-of-the-art. The motivation behind the approach also requires clearer explanation. There is no response from the authors. Therefore, I recommend rejection.

**Justification For Why Not Higher Score:**

N/A

**Justification For Why Not Lower Score:**

N/A

---

### Decision · Program_Chairs · 2024-01-16

Reject